# The Course of SARS-COV2 Infection Was Not Severe in a Crohn’s Patient Who Administered Maintenance Anti-TNF Therapy Overlapping the Early Pre-Symptomatic Period of Infection

**DOI:** 10.3390/antib9030042

**Published:** 2020-08-15

**Authors:** Francis Okeke, Anjali Mone, Arun Swaminath

**Affiliations:** 1DC Gastro Konsult, 11620 Pellicano Dr, Ste B, El Paso, TX 79936, USA; docokeke@gmail.com; 2Lenox Hill Hospital, Northwell Health, 100 E 77th St, 2nd Fl, New York, NY 10075, USA; amone1@northwell.edu

**Keywords:** COVID-19, inflammatory bowel disease (IBD), anti-Tumor Necrosis Factor (TNF) adalimumab, fecal calprotectin

## Abstract

The Inflammatory Bowel Disease (IBD) population, which may require treatment with immunosuppressive medications, may be uniquely vulnerable to COVID-19 infection. In fact, there is some evidence these medications may inhibit the cytokine storm that is theorized to cause a rapid decline seen in COVID-19. In addition, the digestive symptoms of COVID-19 can be difficult to distinguish from the activation of IBD. We present an interesting case of a Crohn’s patient inadvertently administering anti-cytokine therapy during the pre-symptomatic period of COVID-19 infection. Immune suppression during early infection with SARS-COV2 risks a poor immune response to the virus and could theoretically result in a more severe course of infection.

## 1. Introduction

The recently discovered coronavirus (SARS-CoV-2), which causes COVID-19, is now a recognized global pandemic, with the United States reporting the highest number of cases to date. Most IBD patients on treatment are maintained with immunosuppressive agents, and as such are thought to be at an increased risk for infection. Immunosuppression is widely suspected to be a risk factor for severe outcomes/mortality in patients with COVID-19 infection. We present a worst-case scenario of the infection establishing itself at the time of the maintenance administration of anti-TNF-alpha therapy in a patient on combination immunosuppression with methotrexate. 

## 2. Case Report

The patient is a 60-year-old female nurse with a past medical history of autoimmune disease, including rheumatoid arthritis (RA), systemic lupus erythematosus (SLE), and ileocolonic inflammatory Crohn’s disease in endoscopic remission. Both the joint and bowel complaints were controlled for many years on weekly adalimumab 40 mg and methotrexate 12.5 mg. She presented to the emergency department (ED) for the evaluation of one day of persistent fevers and generalized myalgias on March 20th, 2020. Her last dose of adalimumab was administered 5 days prior, while her last dose of methotrexate was administered 7 days prior. 

Her symptoms began the day prior to presentation, with fever (T max 101°F), diffuse myalgias, and fatigue. In the ED, vitals were BP 118/82, HR 91, RR 17, T max 101.1°F, O2 saturation 96% on room air. Labs were notable for normal white count, hemoglobin, and hepatic panel, and negative flu swab. Imaging with chest X-ray was unremarkable. Given her constellation of symptoms, a COVID-19 test was performed. She received 2 liters of normal saline, acetaminophen, and ketorolac with symptomatic improvement with no supplemental oxygen requirement. She was discharged with oseltamivir with a working diagnosis of an influenza-like illness and was requested to quarantine at home until receiving the results of the COVID-19 test, anticipated to take 2–3 days at that time. 

The next day, on 21 March 2020, she noted the onset of gastrointestinal symptoms, including non-bloody, watery diarrhea (5–6 episodes per day, including nocturnal symptoms), nausea with non-bloody, non-bilious emesis, and intermittent para-umbilical abdominal cramping (5/10 in intensity, no radiation, associated with bowel movements and vomiting episodes). She had not experienced a Crohn’s-related flare in over a decade and the symptoms were not consistent with her usual flares. Her fevers persisted and worsened, with a reported T max of 102°F. Three days after the initial ED evaluation, her COVID-19 test returned positive. 

Over the next few days, her symptoms worsened, with new shortness of breath upon exertion, worsening myalgias, fatigue, and generally feeling unwell. She re-presented to the ED for further evaluation on 29 March 2020. At this visit, her vitals were notably worse with T max 101.1°F, BP 152/80, HR 109, RR 17, O2 Sat 99% on ambient air. Labs showed a white blood cell count of 5.29, hemoglobin of 12.9, and normal liver function tests. She was admitted for the management of systemic inflammatory response syndrome (SIRS).

While admitted, she experienced persistent fevers, (T max of 102.4°F), and worsening shortness of breath with desaturations to 87% on room air, which were corrected with 2 L of supplemental oxygen via nasal cannula. A CT scan of the chest (Figure 1) found patchy consolidations in a peripheral distribution scattered throughout the lungs, consistent with known COVID-19 infection, and a small anterior pericardial effusion. A CT scan of the abdomen and pelvis with oral and IV contrast (Figure 2) showed wall thickening in the proximal sigmoid colon, representing colitis. Stool testing for *Clostridium difficile* was negative and a fecal calprotectin test was normal at 19.

The patient was treated with our initial COVID-19 hospitalization protocol during the pandemic, which consisted of a course of hydroxychloroquine 400 mg daily, azithromycin 250 mg daily, and ascorbic acid 500 mg daily for 5 days. The laboratory trend of her inflammatory markers are summarized in Table 1. Supplemental oxygen was required for only two days. She defervesced after day 4 of hospitalization. She was discharged on hospital day 5 and asked to hold her immunosuppression for two weeks. The patient had felt well during her follow up with her gastroenterologist 2 weeks later. 

## 3. Discussion

The COVID-19 pandemic has become a global health crisis, particularly of concern to our IBD patient population, who often require immunosuppressive medications. This case provides some of the granular detail with regard to the timing of infection and the administration of immunosuppressive therapy. Theoretically, immune-modifying treatments place IBD patients at the highest risk for morbidity and mortality from COVID-19 [1]. Patients and providers are concerned about the possibility of an increased risk for acquiring SARS-CoV-2 and the chance for a more severe disease course if infected. In the early stages of the pandemic, available guidelines advised caution when dosing biologics in the setting of a clinically important infection [2,3,4]. Surprisingly, there has not been any significant increase in SARS-CoV-2-driven pneumonia and Acute Respiratory Distress Syndrome (ARDS) documented for immunosuppressed IBD patients [5]. Therefore, the question arises about whether patients treated with cytokine inhibitors are in fact a privileged group, protected from fulminant COVID-19 disease. Our case highlights the course of a patient, treated with a TNF-alpha inhibitor in combination with an immunomodulator, who developed COVID-19. 

Considering the SARS-CoV-2 average incubation period of five days, we estimate our patient administered adalimumab and methotrexate during the pre-symptomatic window, the same time she was likely to have been infected, as shown in the timeline in Figure 3. Although we expected that a patient on combination therapy with increased immunosuppression would rapidly worsen and require intubation, within two days, she improved despite no additional interventions. We hypothesize that the anti-TNF-alpha medication may have interfered with COVID-19’s aberrant immune response, which led to a less severe disease course than expected [6].

It is known that TNF-alpha is necessary for the inflammatory response [7]. COVID-19 pneumonia is characterized by an exaggerated immune response (cytokine storm) with high TNF-alpha levels, as well as other cytokines [8]. Targeted TNF-alpha inhibition may help modulate the immune response and prevent alveolar damage [9,10]. However, there is a tradeoff between suppressing the cytokine storm and mounting an effective response against the virus. The use of immunomodulating medications in COVID-19 still requires further research [5]. A clinical trial is currently evaluating adalimumab for use in treating severe COVID-19 pneumonia, a promising indicator that anti-TNF-alpha medications are, at a minimum, not likely to be harmful to our patients if they are infected [11].

At this point, there are still very limited data on COVID-19 patients with IBD on biologics, and more research is needed. Preliminary data in the surveillance epidemiology of coronavirus (COVID-19) are under research exclusion (SECURE-IBD) [12]. A worldwide database is promising. Of note, the database does not record the times of administration of the medication in comparison to the onset of symptoms. Interestingly, IBD patients with COVID-19 on anti-TNF-alpha therapy do not seem to fare worse than those treated with less immunosuppressive drugs and, furthermore, they seem to do better. 

The second point we wanted to highlight in our patient was the challenge of distinguishing between typical IBD flare symptoms versus COVID-19-related gastrointestinal complaints. In addition to upper respiratory symptoms, a significant number of COVID-19 patients suffer from gastrointestinal symptoms, including loss of appetite, nausea, vomiting, and diarrhea [13]. Furthermore, in a recently published study, almost a third of COVID-19 patients were found to have bowel abnormalities identified on abdominal CT scans. The findings were typical of bowel ischemia thought to be due to vascular injury from small vessel clots [14]. Our patient did have digestive complaints as well as CT imaging showing distal colitis. In this patient population, IBD flares must also remain high on the differential. Interestingly, the fecal calprotectin was normal for our patient, suggesting a different pathophysiology for the intestinal insult other than IBD. 

Calprotectin is a protein found in leukocytes [15,16] and is an acute phase reactant often elevated during inflammation [17]. When there is inflammation in the GI tract, neutrophils migrate to the area and release calprotectin protein, resulting in increased levels detectable in stool. Calprotectin detected in the stool has a direct relationship to mucosal damage from IBD as a consequence of this neutrophil degranulation. This protein is detected in stool using enzyme-linked immunosorbent assays, and studies show that concentrations of calprotectin correlate with the degree of intestinal inflammation [18,19,20]. A positive correlation between CT and endoscopic findings of IBD with fecal calprotectin levels has been documented in the literature [21]. Notably, fecal calprotectin seems to be specific for inflammation related to IBD and selected bacterial infections, not with viral infections [22,23]. In a prior study investigating fecal calprotectin levels in ischemic colitis patients, calprotectin levels could not be correlated with disease activity [24]. Given that the colitis observed in COVID-19 is thought to be ischemic in nature, an elevated fecal calprotectin would likely suggest a cause other than COVID-19.

Differentiating between IBD flares and COVID-19 GI manifestations may help prevent unnecessary colonoscopies and determine the right treatment pathway for patients. The current pandemic of COVID-19 has created a unique risk environment for performing endoscopic procedures. Viral particles have been found in both sputum and stool [25]. Furthermore, studies show virus in the stool beyond three weeks’ post-symptom resolution. Given the limited access to endoscopic evaluation and concerns about limiting provider exposure, the identification of biomarkers to differentiate the cause of symptoms and radiologic abnormalities in patients with underlying IBD will be of great value during this epidemic. It is important to note that the exaggerated inflammatory response to COVID-19 infection makes it difficult to differentiate an IBD flare from COVID-19 GI manifestations, using well-studied inflammatory markers (such as C-reactive protein—CRP, erythrocyte sedimentation rate—ESR) and, as such, we feel it is important to make this determination using a GI-specific marker. We suggest that future studies evaluate fecal calprotectin to identify whether it can be used to distinguish gastrointestinal symptoms and abnormal CT scan findings in COVID-19 infection from IBD flares.

In conclusion, we present an interesting case of an IBD patient on dual immunosuppression, who had a mild course of the COVID-19 disease. We wanted to highlight the point that, despite dosing her anti-TNF-alpha and immunomodulator medications near the time of her exposure, she did not experience a severe disease course. Another important point is the question of trying to discern between COVID-19-related GI symptoms versus IBD flares. We recommend starting with a fecal calprotectin level. It is currently unknown whether COVID-19 affects the course of our IBD patients and whether immunosuppressive treatment affects their course of COVID-19. However, based on outcomes studies, we may be able to allay concerns around immunosuppressed populations during the pandemic.

## Figures and Tables

**Figure 1 antibodies-09-00042-f001:**
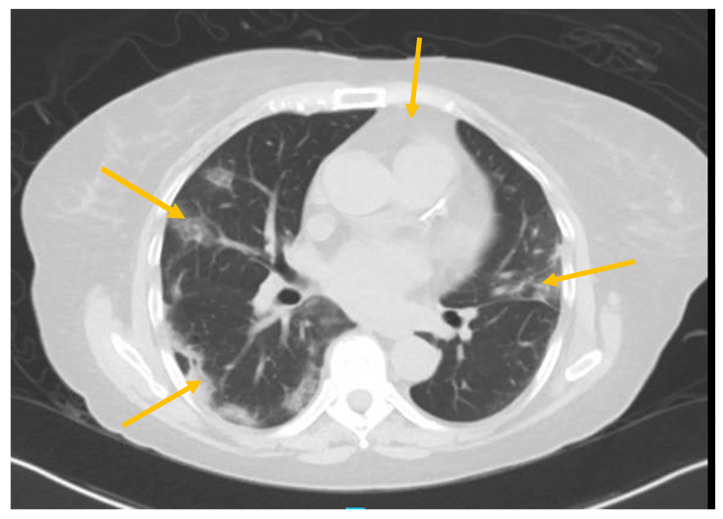
CT of chest showing patchy consolidations in peripheral distribution and pericardial effusion (arrows).

**Figure 2 antibodies-09-00042-f002:**
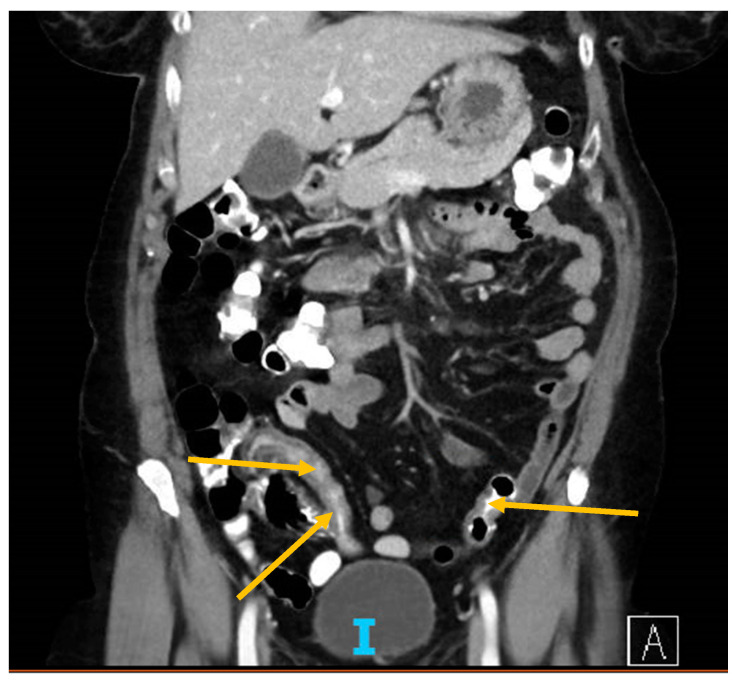
CT of abdomen and pelvis showing sigmoid colon thickening (arrows).

**Figure 3 antibodies-09-00042-f003:**
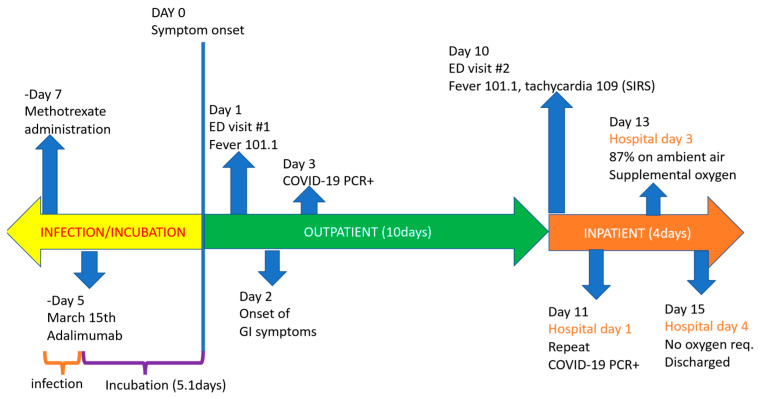
Timeline.

**Table 1 antibodies-09-00042-t001:** Vital/laboratory values/trend.

Symptom/Date	03/29/2020	03/30/2020	03/31/2020	04/01/2020	04/02/2020	04/03/2020
Temperature (T max °F)	101.1	100.1	101.8	102.4	100	99
Diarrhea	+	+	+	+	+	-
Nausea/Emesis	+	+	-	-	-	-
CRP	3.96	4.66	3.80	3.06	3.73	2.21
D-dimer	199	222		<150	203	191
Ferritin	602	680	811	766	884	792
Fecal calprotectin						19
Alkaline phosphatase	62	61	76	78	82	83
Aspartate aminotransferase	39	44	65	80	95	92
Alanine aminotransferase	34	43	64	80	113	118

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
