# Peer review of "The Course of SARS-COV2 Infection Was Not Severe in a Crohn’s Patient Who Administered Maintenance Anti-TNF Therapy Overlapping the Early Pre-Symptomatic Period of Infection"

_2073-4468, 2020, doi:10.3390/antib9030042_

Round 1

Reviewer 1 Report

The case report submitted by Okeke et al. demonstrated the situation of the course of SARS-COV2 infection when overlaps with the administration of anti-TNF-alpha therapy in IBD patient, which is really an interesting and worth studying literature during the crisis situation of COVID-19 pandemics worldwide. The article has outstanding merits and interests in its own field. The manuscript has been well-written and presented in nice shape.

However, here is few minor concerns-

  1. The authors provided the detail clinical records and follow up management for COVID-19 patient until day 15 (the discharged day from hospital) from the onset of symptoms. She was also tested positive at Day 11 for COVID-19. Then, this patient was discharged from hospital based on basically no requirement of supplementary O2 but without a negative status COVID-19 test. Did authors have further follow up or other information weather the patient was completely recovered from COVID-19 infection and how long was taken for full recovery? Or what was the situation after 2 weeks of holding her immunosuppression medication?
  2. Anti-TNF-alpha medication could be a good alternative treatment strategy in COVID-19 infection, but authors need to be clarify in more cases in future.
  3. Anti-TNF could be replaced by Anti-TNF-alpha therapy…..in the title of article and throughout the manuscript.

Author Response

  1. The authors provided the detail clinical records and follow up management for COVID-19 patient until day 15 (the discharged day from hospital) from the onset of symptoms. She was also tested positive at Day 11 for COVID-19. Then, this patient was discharged from hospital based on basically no requirement of supplementary O2 but without a negative status COVID-19 test. Did authors have further follow up or other information weather the patient was completely recovered from COVID-19 infection and how long was taken for full recovery? Or what was the situation after 2 weeks of holding her immunosuppression medication?

ANSWER –

This is an important question that has not been settled in the scientific literature.

Our institutions’ protocol at the time and currently do not require a negative COVID-19 test prior to discharge from the hospital to home (unlike for skilled nursing facilities).  Positive pcr tests are common long after resolution of symptoms and it’s not clear whether this reflects live virus or an overly sensitive test with no clinical relevance, though this issue has not been settled in the scientific literature.  Most patients with COVID-19 are managed with symptomatic treatment and self-quarantine unless they need supplemental oxygen and further hospital management. 

As recommended by the reviewer, we had added the following to the case report to clarify the status of the patient during a longer window.

The patient did not return to the hospital for readmission in 2 weeks and on follow up with her primary gastroenterologist she is doing well since discharge from the hospital

  1. Anti-TNF-alpha medication could be a good alternative treatment strategy in COVID-19 infection, but authors need to be clarify in more cases in future.

ANSWER –

We agree with this suggestion and do mention that further studies are needed to prove or disprove this possibility.

  1. Anti-TNF could be replaced by Anti-TNF-alpha therapy…..in the title of article and throughout the manuscript.

ANSWER –

We agree with the reviewer and have made this change.

Reviewer 2 Report

The submitted manuscript by Okeke et al is a nice description of an interesting case of a 60-year-old IBD patient on immunosuppression that demonstrated a mild course of the COVID-19 disease.

Comments

  1. It would be good to expand the introduction a little bit by including some information about IBD. The authors also need to show why this represents a worse case scenario as mentioned in the introduction.
  2. Arrows should be included on the figures to indicate specific portions of importance. Additionally, control figures, either from old images from the same patient or from another healthy patient, should be included to explain the observations on figures 1 and 2.
  3. It would be good to have the grammar of the manuscript reviewed for correct English (for example, 101°F not 101F; 250 mg not 250mg; Figure 1 not Figure1).
  4. I would suggest that the title of the report be based on the conclusions instead of making it a question.

Author Response

Comments

  1. It would be good to expand the introduction a little bit by including some information about IBD. The authors also need to show why this represents a worse case scenario as mentioned in the introduction.

ANSWER – We agree and have changed the title to:

The course of  SARS-COV2 Infection was not severe in a Crohn’s patient who administered maintenance Anti-TNF Therapy overlapping the early pre symptomatic period of infection.

In addition, we have added to the following sentence to the abstract to further emphasize the “why” of worst case scenario.

Immune suppression during early infection with SARS-COV2 risks a poor immune response to the virus and could theoretically result in a more severe course of infection. 

  1. Arrows should be included on the figures to indicate specific portions of importance. Additionally, control figures, either from old images from the same patient or from another healthy patient, should be included to explain the observations on figures 1 and 2.

ANSWER – We agree and have added arrows to the abnormal areas.  As the patient did not have a CT prior to presentation, we do not have a preinfection scan to provide, though her lack of prior pulmonary complaints would suggest she would have a  normal CT scan prior to infection.

  1. It would be good to have the grammar of the manuscript reviewed for correct English (for example, 101°F not 101F; 250 mg not 250mg; Figure 1 not Figure1).

ANSWER –

Changes have been made to the necessary sections

  1. I would suggest that the title of the report be based on the conclusions instead of making it a question.

ANSWER – We agree and have changed the title to “The course of  SARS-COV2 Infection was not severe in a Crohn’s patient who administered maintenance Anti-TNF Therapy overlapping the early pre symptomatic period of infection.”